# Detection of Ciguatoxins and Tetrodotoxins in Seafood with Biosensors and Other Smart Bioanalytical Systems

**DOI:** 10.3390/foods12102043

**Published:** 2023-05-18

**Authors:** Jaume Reverté, Mounira Alkassar, Jorge Diogène, Mònica Campàs

**Affiliations:** Institut de Recerca i Tecnologia Agroalimentàries (IRTA), Ctra. Poble Nou km 5.5, 43540 La Ràpita, Spain; jaume.reverte@irta.cat (J.R.); mounira.alkassar@irta.cat (M.A.); jorge.diogene@irta.cat (J.D.)

**Keywords:** marine toxins, biosensors, bioanalytical assays, fish, shellfish, food safety

## Abstract

The emergence of marine toxins such as ciguatoxins (CTXs) and tetrodotoxins (TTXs) in non-endemic regions may pose a serious food safety threat and public health concern if proper control measures are not applied. This article provides an overview of the main biorecognition molecules used for the detection of CTXs and TTXs and the different assay configurations and transduction strategies explored in the development of biosensors and other biotechnological tools for these marine toxins. The advantages and limitations of the systems based on cells, receptors, antibodies, and aptamers are described, and new challenges in marine toxin detection are identified. The validation of these smart bioanalytical systems through analysis of samples and comparison with other techniques is also rationally discussed. These tools have already been demonstrated to be useful in the detection and quantification of CTXs and TTXs, and are, therefore, highly promising for their implementation in research activities and monitoring programs.

## 1. Introduction

Marine toxins are a diverse group of complex chemical compounds produced as secondary metabolites by a variety of toxic microalgae and in some cases by bacteria or other organisms. Some marine toxins may accumulate in shellfish or fish and be transferred through food webs, reaching consumers and causing poisoning and even death [1]. Their harmful effects on human health have forced the development and implementation of regulations to set their maximum allowed levels in seafood as well as the methodologies to be used for their quantification in samples. However, several groups of marine toxins, in particular ciguatoxins (CTXs) and tetrodotoxins (TTXs), have recently emerged in Europe and, therefore, still need to be further assessed and regulated [2].

Ciguatoxins are potent lipophilic marine toxins produced by microalgae of the genera *Gambierdiscus* and *Fukuyoa* [3]. Historically, CTXs were classified into three groups according to their geographical origin: Pacific (P-CTXs), Caribbean (C-CTXs), and Indian CTXs (I-CTXs). Advancements on their structure elucidations have allowed their classification into three groups according to their chemical structure: CTX4A analogues, CTX3C analogues, and Caribbean CTXs (C-CTX1 and C-CTX2) [4]. At a cellular level, CTXs block the voltage-gated sodium channels (VGSCs) in an open position by binding to site 5 on the α subunit and increasing the sodium concentration inside the cells [5]. Ciguatoxins, being heat-stable and with no odor or taste, are responsible for the most prevalent seafood-borne disease, called ciguatera poisoning (CP). This poisoning causes acute effects including digestive, cardiovascular, and some characteristic neurologic symptoms such as cold dysesthesia, described as painful tingling, burning, smarting, “electric shock”, or “dry ice” sensations in the skin when it comes into contact with normally innocuous cold [6]. Ciguatera poisoning also causes chronic neurological effects that may last for months or even years. Ciguatera poisoning is suggested to affect between 10,000 and 500,000 people each year worldwide [7]. Historically, CP was typical of tropical and subtropical regions. However, in recent years, cases of CP in more temperate regions have been reported. In Europe, several CP outbreaks have occurred in the Canary Islands (at least 21 episodes since 2008) [8]. Ciguatoxins have also been detected in fish from other areas of Macaronesia, such as Azores and Madeira (Portugal) [9,10,11]. Several species of the genus *Gambierdiscus* have been recorded in the Canary Islands [12,13,14,15] and Madeira [16,17], but also in the Mediterranean Sea [18,19,20]. Regarding regulations, the US Food and Drug Administration (FDA) proposed a guidance level of 0.01 µg P-CTX-1B equivalents/kg of tissue or 0.1 µg C-CTX-1 equivalents/kg of tissue [21]. In Europe, the legislation determines that fishery products containing CTXs must not enter the market [22].

Tetrodotoxin is a potent neurotoxin, with a low molecular weight, discovered in pufferfish ovaries by Tahara and Hirata in 1909 [23]. Although TTX is well known for being present in some species of pufferfish, it has also been found in gastropods, newts, crabs, frogs, sea slugs, star fishes, blue-ringed octopuses, ribbon worms, and bacteria [24]. To date, more than 30 TTX analogues have been reported [25]. At a cellular level, TTXs have a mechanism of action opposite to that of CTXs. Tetrodotoxins block the VGSCs in a close position by binding to site 1 on the α subunit and inhibit the influx of sodium ions through the cell membrane [26]. In humans, this effect can result in perioral numbness, incoordination, respiratory paralysis, gastrointestinal effects, and, in severe cases, death may occur due to respiratory and/or heart failure [27]. In recent years, intoxications due to the consumption of toxic pufferfish have been described in Mediterranean countries. Additionally, TTXs have also been found in shellfish from several European countries, including the United Kingdom [28], Greece [29], the Netherlands [30], Portugal [31], Spain [32,33], Italy [34,35], and France [36,37]. Regarding regulations, the Japanese government has established a regulatory limit of 2 mg TTX equivalents/kg of tissue in pufferfish [38]. In Europe, fishery products derived from the Tetraodontidae family cannot be marketed [39]. Regarding its presence in shellfish, the European Food Safety Authority (EFSA) proposed a guideline value of at 44 µg TTX equivalents/kg of shellfish meat, since it does not result in adverse effects in humans [40], but no regulation level has been established yet.

The emergence of marine toxins such as CTXs and TTXs in non-endemic regions may pose a serious food safety threat and a public health concern if proper control measures are not applied. In this context, the development of highly reliable and efficient analytical methods for the detection of these toxins is essential for seafood management and food-borne poisoning prevention. In the last 15 years, the number of publications with regards to new analytical methods for marine toxins has reached 200 per year, and the citations have increased from 5000 to more than 11,000. Regarding traditional methods, animal bioassays, such as the mouse bioassay (MBA), were the first methods used for the detection of marine neurotoxins [41]. MBA was and still is very useful because it provides a composite toxicological response and, therefore, helps to understand and manage poisoning episodes. However, MBA, although very cheap, has specificity and ethical concerns. Although still in use in some countries, it is banned in some others or restricted to specific cases, when no alternative exists. Instrumental analysis methods, such as ultra-performance liquid chromatography coupled to fluorescence detection (UPLC-FLD), liquid chromatography coupled to tandem mass spectrometry (LC-MS/MS), or liquid chromatography coupled to high-resolution mass spectrometry (LC-HRMS), have been developed for the detection of these toxins [42]. Those techniques coupled to mass spectrometry are highly sensitive, precise, and provide unequivocal confirmation of the presence of these toxins (and even different toxin analogs) in samples. However, they may suffer from limitations such as expensive equipment, the need for skilled operators, and long analysis times. Additionally, they do not provide toxicological information.

Alternative methods for the detection of neurotoxins include cell-based assays (CBAs), receptor binding assays (RBAs), immunoassays, and aptamer-based assays. Some of these bioanalytical assays have been formatted into biosensor platforms, with added advantages in terms of ease of use, miniaturization, automatization, and portability. Biosensors and other bioanalytical tools for the detection and quantification of marine toxins are gaining interest and could be implemented in research activities and monitoring programs after development and proper validation.

## 2. Biorecognition Elements

Biosensors are analytical systems with a biorecognition element and a transducer in intimate contact. The biorecognition element specifically recognizes a target analyte and the transducer converts the biorecognition event into a signal that can be measured and that is proportional (or inversely proportional) to the analyte concentration. The inherent properties of the biorecognition element and the assay configuration play a key role in the analytical performance parameters of the biosensor, such as sensitivity or specificity. Therefore, the biorecognition element choice and the assay design need to be carefully and rationally assessed on a case-by-case basis.

### 2.1. Cells

Historically, in toxicology, the detection of an analyte was based on the observation of its toxic effect on animals. For example, in MBAs, contaminated seafood extracts were administrated orally or intraperitoneally to mice and the intoxication evolution, sometimes until animal death, was monitored [43]. The use of animals for toxin screening has limited detection capabilities, such as lack of specificity and huge variability among analyses, in addition to ethical concerns. For this reason, their use has been limited to, for example, the analysis of samples with unknown toxicities [44], the evaluation of alterations produced by toxins at anatomic or organic level [45], and the description of in vivo symptoms that may help to characterize their mechanism of action or the determination of toxin lethal doses [46,47].

Cell-based assays, with cells as biorecognition elements, have been proposed as an alternative approach to assess toxicity of samples without the need for live animals [48]. With well-established cell-lines, in vitro CBA can provide more uniform responses than in vivo MBA, which suffers from higher variability among mice. Initially, the detection of toxins with CBA was based on the visual inspection of morphological or structural changes of cells exposed to toxins [49], which required trained personnel and was operator dependent and time consuming. To overcome these limitations and provide accurate quantifications, colorimetry was used for cell viability evaluation [50,51]. The most commonly used colorimetric CBA is based on the ability of metabolically active cells to reduce a tetrazolium salt, usually (3-(4,5-dimethylthiazol-2-yl)-2,5-diphenyltetrazolium bromide) (MTT), to the blue-colored formazan precipitate, which, once dissolved, turns purple and can be measured by spectrophotometry, providing a quantification of cell viability [52].

As explained above, CTXs and TTXs are VGSC-specific neurotoxins. This ionic channel is highly expressed in excitable tissues such as the nervous system. In this sense, the use of primary cultures, freshly initiated from animal nervous tissues, for the study of CTXs and TTXs has been extensively explored in the literature [53,54,55,56]. However, their applicability to routine toxin analysis is compromised by the high dependence on live animals as a source of cells and the difficulties associated with culturing this type of cells. On the contrary, established immortal lines, like Neuro-2A cells obtained from murine neuroblastoma, are easier to culture and have been extensively used for the analysis of marine neurotoxins. However, they usually express fewer ionic channels than primary cultures [57]. In fact, Neuro-2A cells are not very sensitive to CTXs or TTXs and require a pre-treatment with two auxiliary drugs, ouabain (O) and veratridine (V), to properly respond to these toxins [49]. As demonstrated in Figure 1, V blocks VGSCs in an open state (increasing the intracellular sodium levels), whereas O blocks the Na^+^/K^+^ ATPase pump (one of the main cellular mechanisms of sodium excretion). The combined effect of O/V at moderate concentrations reduces cell viability but maintains a proportion of cells that are more sensitive to toxins. Ciguatoxins, which also open VGSCs, enhance the effect of V producing a dose-response reduction on cell viability. Instead, TTXs block VGSCs in a close state, causing an effect antagonistic of V, increasing cell viability. When analyzing samples, CBAs provide a composite toxicological response, which includes the effect of all toxic congeners, although these may have different toxicity. Toxicity equivalency factors (TEFs) are useful to express the toxicity of a congener relative to a reference compound. It is important to mention that since detection is based on a toxicological evaluation, toxins with similar modes of action, such as CTXs and brevetoxins (PbTXs) or TTXs and saxitoxin (STX) and its congeners, cannot be differentiated.

### 2.2. Receptors

In RBAs, detection is based on the interaction of toxins with cellular membrane receptors. These biorecognition elements are usually embedded into membrane fragments (synaptosomes) isolated from whole cells. Like CBAs, RBAs provide a composite response from a sample, as they are able to detect several toxin analogues or related compounds that interact with the receptors (although with different affinities). Since the toxin-receptor interaction does not induce a toxic effect in RBAs, this method cannot be strictly considered a toxicological approach. Nevertheless, it is clear that in toxicological models, the toxic response starts with this interaction. Therefore, several scientists consider RBAs to be toxicological and others structural.

As explained above, the expression of molecular receptors depends on the tissue or cell lineage. Therefore, the origin of the synaptosomes may affect the performance of the RBA. In mammals, at least nine subtypes of VGSCs are expressed (Na_v_1.1–Na_v_1.9). Typically, Na_v_1.1–1.3 and Na_v_1.6 are more highly expressed in the central nervous system, Na_v_1.7–1.9 in the peripheral nervous system, and Na_v_1.4 and Na_v_1.5 in muscle [58]. All subtypes share the same tridimensional structure and functionality. However, the affinity towards toxins is different due to small differences on the peptide sequence identity [59]. For example, Na_v_1.1–1.4, Na_v_1.6 and Nav1.7 are sensitive to TTX, whereas Na_v_1.5, Na_v_1.8, and Na_v_1.9 are not [60]. Regarding CTXs, all VGSC subtypes demonstrated sensitivity to CTX1B, the most toxic CTX congener [61]. As previously mentioned, the VGSC expression yield in Neuro-2A cells is much lower than in neuronal tissues, and it depends on the culturing conditions, the growing phase, and the degree of cellular differentiation [57]. Therefore, most synaptosomes used in RBAs are obtained from primary cultures initiated from animal tissues instead of from immortal cell lines. An advantage is that batches of isolated receptors may be cryopreserved and can be used at convenience.

Regarding the format, most RBAs are based on a competition between the toxin present in the sample and a labeled ligand for binding to the receptor [2]. In a competition format, the signal is inversely proportional to the toxin content. In other words, the lower the toxin content is in the sample, the more the labeled ligand is bound to the receptor. On the contrary, in samples with high toxin amounts, lower amounts of the labeled ligand are bound to the receptor, giving rise to low signals (Figure 2). Originally, the competition was performed in suspension. However, to avoid the filtration or centrifugation steps required and to simplify the assay, synaptosome immobilization has been explored. Additionally, radioactive isotopes are usually used as labels. Nevertheless, to avoid the use of hazardous compounds, alternative labels for chemiluminescent, fluorescent, or colorimetric detection have been explored.

### 2.3. Antibodies

Immunoassays are a type of structural recognition approach that, in contrast with CBAs, do not provide information about sample toxicity. In immunoassays, recognition is based on antibodies, which are biological agents produced by immune cells in response to external antigens (e.g., toxins). The production of antibodies usually starts by immunization of mice or rats with the target analyte [62]. Sometimes, particularly with small antigens, the target analyte is conjugated to a carrier protein to increase its antigenicity, stimulate the immune system, and enhance antibody production. This strategy was explored by Kawatsu and co-workers, who produced an antibody against TTX using a bovine serum albumin (BSA)-TTX conjugate [63]. Generally, target molecules have different antigenic regions (epitopes) in their structures. Each immune cell recognizes one single epitope and, consequently, the antibodies that each immune cell produces are epitope specific. After immunization, the animal serum displays a mixture of antibodies that recognize the antigen at different epitopes, a mixture known as polyclonal antibodies (pAb). Each one of these antibodies comes from a different immune cell.

Although pAbs have been used for the development of immunoassays, monoclonal antibodies (mAbs) are usually preferred. This type of antibody is produced from individual antibody-producing cells obtained from mice and subsequently fused with immortal cancer cells from myeloma, forming hybridomas [64]. Each hybridoma is selected and individually cultured, producing antibodies that recognize the antigen at a specific epitope. Even though mAbs are highly specific, sometimes they can also recognize toxin analogues or other related compounds if they share the same epitope. This cross-reactivity could be beneficial or detrimental depending on the purpose of the analysis [2]. In the development of analytical approaches for the detection of families of toxins, ideally the antibody should be able to recognize all toxin analogues that are toxic and, consequently, have a role in poisoning.

In addition to classical antibody producing approaches, biotechnological advances in molecular biology have led to the production of recombinant antibodies using new expression systems [65]. These recombinant antibodies have demonstrated good biorecognition abilities and have already been used in the development of analytical systems.

### 2.4. Aptamers

Aptamers are oligonucleotide sequences that can be used as biorecognition elements in analytical systems due to their ability to specifically bind to some targets. Despite being oligonucleotides, the target recognition is not based on hybridization with the complementary sequence, but on the tridimensional folded configuration that they adopt [66]. Like antibodies, aptamers do not provide information about the toxicity of a sample. For that reason, aptamer-based analytical systems are considered structural approaches.

Aptamers are obtained from oligonucleotide libraries by a selection process known as SELEX (Systematic Evolution of Ligands by Exponential Enrichment) [67]. In the classical SELEX configuration, sequences from an oligonucleotide library are exposed to the target of interest, which has been previously immobilized on a support. After incubation, sequences that have not interacted with the target are discarded, whereas those bound to the target will be eluted and amplified. The product of amplification will be exposed again to the target, and this cycle will be repeated several times. At the end of this enrichment process, one or a few aptamers with high affinity against the target will be obtained. Different variations of SELEX have been explored to improve the production of aptamers and their subsequent use [68]. For example, in counter SELEXs, additional exposure steps are introduced in the process using structurally similar compounds (e.g., other toxins that could co-exist in the sample) in order to discard those aptamers able to cross-react with compounds other than the target of interest [68]. For small-size targets, such as TTXs, a capture SELEX is preferred, which is based on the immobilization of the library of sequences instead of the target. The incubation of TTX in solution instead of immobilized TTX favors its interaction with the oligonucleotides [69]. Unfortunately, up to now, no aptamers against CTXs have been described, with the required target amount being one of the limitations.

## 3. Biosensors and Other Smart Bioanalytical Systems

In the development of a biosensor, different strategies for immobilization and transduction can be envisaged, and the choice will depend on the desired performance characteristics and prototype format. In this section, we describe the main biosensors and smart bioanalytical systems developed for the detection of CTXs and TTXs.

### 3.1. Cell-Based Biosensors

In cell-based biosensors (CBBs), the physiological disorders of cells produced by toxins are recorded by electrodes or other transducers, and subsequently transformed into measurable signals that are processed to an output that describes the toxicological potential of the sample. In principle, CBBs have added advantages in comparison with CBAs, such as shorter analysis time, miniaturization, and automatization. However, the number of CBBs developed until now for detection of neurotoxins such as CTXs or TTXs is relatively low, compared with CBAs.

The culture of living cells on transducers is a critical issue in the development of a CBB, since the immobilization of cells should not affect their biological functions [70]. For this reason, an immobilization based on physical adsorption mediated by the adherence properties of the cells is desirable to guarantee the viability of cells and their proliferation. The adsorption of cells on sensing materials can be enhanced through the modification of these surfaces with components that emulate the extracellular matrix environment [71]. For example, polymers such as polyaniline, poly-L-lysine, and poly(3,4-ethylenedioxythiophene) have been proven to be effective for the adhesion of Neuro-2A cells on carbon electrodes [72]. Another factor that needs to be considered in the immobilization of cells is signal transduction. For example, hydrogels provide a 3D environment suitable for the immobilization of cells on electrodes [73]. However, their use may reduce electron transfer, hindering detection. For that reason, if possible, intimate contact between the cell and the transducer must always be sought.

In a CBB, several parameters can be recorded to evaluate the toxic activity of CTXs and TTXs. Since the target of these toxins are the VGSCs, most CBBs are based on the evaluation of cellular membrane activity, whose change in functionality is one of the first responses to the effect of toxins. Electrophysiology is the discipline that studies changes in the electrical properties of excitable cells (e.g., neurons) caused by the alteration of ionic flow through the cellular membrane. When resting, excitable cells present a heterogeneous ion distribution between the intracellular and extracellular environment, having more positive charges (Na^+^/K^+^) outside than inside the cell. As a consequence of this gradient of charges, a resting membrane potential is originated. The application of an electrical current produces excitation of these cells by activation of the voltage-gated ionic channels. In a normal physiological situation, this activation causes the influx of ions into the cell, changing the potential of the membrane [74]. These changes in the membrane potential are transferred to the immediate surrounding fluid of the cells and can be measured using electrodes. This is the detection principle of the biosensing platforms based on electrophysiology for CTXs and TTXs.

One of the techniques to perform electrophysiological studies is patch clamp. The classical patch clamp technology, also known as cell-attached patch clamp, is based on the immobilization of single cells by suction at the end of a micropipette filled up with an electrolyte solution. An electrode is immersed into the electrolyte solution (inside the pipette) and located close to the cell membrane, whereas another electrode is immersed in the media surrounding the cells (outside the pipette) to be used as a reference electrode [75]. The pipette limits a specific portion of the cellular membrane, which only contains one or a few ionic channels [76]. The toxic sample is added inside the pipette, and therefore the toxin comes in contact with these few ionic channels present at the end of the pipette (Figure 3A). Several variations of patch clamp have been explored with different analytical purposes. In a whole-cell patch clamp configuration, in contrast with the classical patch clamp approach, the effect of the toxins is evaluated in the entire cell membrane and, therefore, the signal recording requires the introduction of the electrode inside the cell [77]. To provide access to the cytoplasm, higher suction intensity is applied to induce a rupture on the membrane while still maintaining the seal of the cell. In this type of configuration, the toxins are not added inside the pipette but in the surrounding cell media, instead (Figure 3B). This whole-cell patch clamp configuration has been used to evaluate the properties of TTX as a drug for the treatment of arrhythmias or other disorders [78]. The cell immobilization on the micropipette is a key factor, since a good seal is necessary to achieve an electrochemical signal. The introduction of automated systems for cell manipulation has simplified the process and reduced variability among analyses [79]. Raposo-Garcia and co-workers explored the automated patch clamp strategy to calculate the TEFs of different CTX congeners and related compounds associated with CP, using a human embryonic kidney cell line (HEK293) expressing the human Na_v_1.6 VGSC [80].

Microelectrode arrays (MEAs) have also been used for the electrophysiological evaluation of neuronal cells in response to toxins. This type of biosensing device includes multiple electrodes placed at the bottom of the wells in tight contact with the cells for signal recording (Figure 3C). The main characteristic of this system is that it allows the electrophysiological evaluation of a cell culture instead of individual cells, being a smart approach to assess neuronal network responses to toxins or other drugs. The use of MEAs as sensing supports for the detection of TTXs [56,81,82] and CTXs [83] has been explored. Nicolas and co-workers have additionally analyzed extracts of mussels artificially contaminated with TTX, and no matrix interferences were observed [83]. Even though some MEA configurations have been able to detect toxins at very low contents, more studies about matrix interferences are needed to validate their applicability to seafood analysis.

Another type of CBB is the strategy used by Alkassar and co-authors, where cell viability is recorded with electrochemical techniques [72]. As explained above, in metabolic active cells, the MTT dye is transformed to a formazan product with redox properties. In this work, this formazan salt precipitates on the electrode, enabling electrochemical measurements. The more metabolically active the cells are, the more formazan salt will be produced, and higher electrochemical signals will be obtained (Figure 3D). The approach was tested with naturally contaminated fish samples, *Seriola dumerili* containing CTXs and *Lagocephalus sceleratus* containing TTXs, providing successful screening results.

### 3.2. Receptor Binding Biosensors

The use of RBA for the detection of marine toxins has been explored during the last decades. The International Atomic Energy Agency (IAEA) has elaborated a manual of RBA methods for different groups of toxins, including CTXs [84]. This methodology was adapted by Diaz-Asencio and co-workers for the development of an RBA for the detection of CTXs using radiolabeled PbTX with tritium (^3^H-PbTx-3). This approach was applied to the analysis of naturally contaminated fishes from Cuba [85]. Darius and collaborators developed a similar RBA for the analysis of CTXs, but besides of fish, they also applied it to the analysis of *Gambierdiscus* cells [86]. Regarding TTXs, Rivera and co-authors described an RBA for TTXs using tritiated STX as a radiolabeled tracer [87]. In this case, the RBA was applied to the evaluation of the ability of an anti-TTX antibody to neutralize TTX.

As explained above, despite having potential as an analytical approach, the use of radioisotopes as labels may be hindering the applicability of these RBAs. For that reason, there is a tendency towards the use of alternative labels in the production of tracers (e.g., fluorophores). Hardison and co-workers developed a fluorescent RBA for the detection of CTXs in fish also using PbTX as a tracer but, in this case, labeled with a fluorescent dye [88]. They had to clean the samples with chemical procedures to remove the matrix that was quenching the fluorescence signal. After cleaning, no matrix effects were observed, and good analytical parameters were obtained. This approach led to the development of a commercial kit for the detection of CTXs and PbTX in fish, supplied by the company SeaTox Research Inc. [89]. Unfortunately, no receptor-based biosensors (RBBs) have been described until now.

### 3.3. Immunosensors

Immunosensors are bioanalytical systems based on the interaction between antibodies and antigens, like immunoassays. The shift from a colorimetric immunoassay to an electrochemical immunosensor requires the replacement of microtiter plates with electrodes and the rational design of an electrochemical transduction strategy. Nevertheless, colorimetric immunoassays are usually previously developed to assess the proper antibody-antigen recognition. The most common immunoassay format used in toxin detection is the enzyme-linked immunosorbent assay (ELISA), characterized by the use of enzyme labels for signal development. ELISAs can be direct or indirect depending on whether the antibody is labeled with the enzyme (direct) or a secondary enzyme-labeled antibody is needed for signal development (indirect). The use of an indirect format increases the assay time, but it may be convenient because secondary labeled antibodies are commercially available. On the contrary, the modification of antibodies with labels may not be straightforward. Nevertheless, the use of a direct ELISA reduces the analysis time and, sometimes, may increase the sensitivity. Although alkaline phosphatase (ALP) has been used as an enzyme label, horseradish peroxidase (HRP) is most common due to its high catalytic activity (high response in short time). In both cases, electrochemical transduction strategies can be sought to develop the corresponding electrochemical biosensors.

The size and structure of the target analyte play an important role in immunosensing configuration [90]. With low-molecular weight antigens, such as TTXs, competitive formats are usually required. In this format, the target present in the sample competes with the target previously immobilized on the support for binding with the antibody. The more toxin is present in the sample, the less antibody will bind to the immobilized toxin and, consequently, a lower signal will be generated (Figure 4A). The way the target is immobilized on the support can substantially affect the detection capabilities of the whole assay, since the antigen needs to be sufficiently exposed to be recognized by the antibody. The adsorption of target-BSA conjugates on microtiter plates was the first strategy used in the development of a competitive ELISA for TTXs. Although effective, and also applied in the development of electrochemical biosensors [91], this strategy may suffer from high non-specific signals, and the random orientation of the immobilized toxin may affect the sensitivity of the assay or biosensor. For this reason, alternative strategies for oriented and spaced toxin immobilization have been explored to favor its exposure.

In this direction, Reverté and colleagues immobilized TTX on maleimide-coated microtiter plates through a self-assembled monolayer (SAM) of carboxylated polyethylene glycol dithiols [92]. The immunoassay was applied to the analysis of pufferfish of the species *L. sceleratus* from Greece, being able to detect TTXs at levels as low as 0.23 mg TTX/kg, which is almost 10-fold lower than the level that the Japanese regulation considers safe for consumption. This immunoassay was shifted to gold electrode arrays for the development of an electrochemical immunosensor [93]. The SAM not only provided an ordered and oriented TTX immobilization, but also facilitated the electrochemical redox mediator to reach the electrode surface. As a result, the limit of detection decreased down to 0.07 mg TTX/kg. Both the assay and the biosensor were applied to the analysis of a pufferfish individual of the species *L. sceleratus* caught in Dènia (Spain) in 2014 [94].

The colorimetric immunoassay developed for pufferfish was evaluated for its applicability to the analysis of shellfish [95]. In this case, dithiols were replaced by cysteamine to simplify the TTX immobilization and to reduce the analysis time and cost. The immunoassay demonstrated strong matrix effects. For that reason, extracts had to be cleaned up with solid-phase extraction (SPE) columns. SPE filtration decreased TTX recovery, but the system was still able to detect 20 µg TTX/kg in oysters and around 50 µg TTX/kg in mussels. This immunoassay was applied to the analysis of naturally contaminated oyster and mussel samples from the Netherlands. Taking into consideration the EFSA guideline value of 44 µg TTX equivalents/kg of shellfish, some improvements were still necessary to be able to attain these low TTXs contents in mussels. Magnetic beads (MBs) have been widely used as immobilization supports in colorimetric assays and electrochemical biosensors. The use of MBs is convenient because they improve assay kinetics, increase washing efficiency, and reduce matrix effects [96]. Campàs and co-workers demonstrated that constructing the same immunoassay on MBs instead of on microtiter plates improved the performance of the assay [97]. The system was able to detect TTX at levels as low as 1 µg TTX/kg of oysters and razor clams and 3.3 µg TTX/kg of mussels (well below the previous results and the EFSA recommended limit) and was applied to the analysis of the same samples from the Netherlands. Leonardo and collaborators explored the same configuration on MBs for the development of an electrochemical immunosensor [98]. In this case, MBs were immobilized on carbon electrodes using magnets and electrochemical signals from the enzyme label were recorded. The system was successfully applied to the analysis of pufferfish juveniles from Greece, which contained high TTX contents.

In the case of CTXs, most immunoassays and immunosensors developed until now are based on a sandwich configuration (i.e., the target detection is achieved with two antibodies that recognize the antigen at different epitopes). The capture antibody is immobilized on the support, and the reporter antibody is added once the target analyte has been captured. In contrast with competitive ELISA, the signals obtained in a sandwich configuration are directly proportional to the amount of toxin present in the sample (Figure 4B). Tsumuraya and Hirama produced antibodies for the detection of the four main P-CTX congeners (CTX1B, 54-deoxyCTX1B, CTX3C, and 51-hydroxyCTX3C), using synthetic fragments of CTXs (haptens) conjugated to carriers for the immunization of animals instead of CTXs standards [99]. The left wing of CTX1B and 54-deoxyCTX1B is recognized by the 3G8 mAb [100], whereas the left wing of the CTX3C and 51-hidroxyCTX3C is recognized by the 10C9 mAb [101]. The right wing of all four congeners is recognized by the 8H4 mAb [102]. The combination of these antibodies allowed the development of sandwich immunoassays, where the 3G8 and 10C9 mAbs were used as capture antibodies and 8H4 mAb was labeled with enzymes to be used as a reporter.

The high specificity and sensitivity of the mAbs and their good performance on sandwich immunoassays has led to the development of electrochemical biosensors for the analysis of CTXs. The first approach used MBs as an immobilization support for both capture mAbs [103]. After exposing the sample, the reporter antibody, modified with biotin, was added to the system and subsequently incubated with polyHRP-streptavidin for signal recording. The biosensor was able to detect CTXs contents in fish extracts as low as 0.01 µg CTX1B equivalents/kg (FDA regulation level) and was successfully applied to the analysis of fishes from La Réunion and Maurice. Gaiani and co-authors used this biosensor to analyze cultures of microalgae cells from the genera *Gambierdiscus* and *Fukuyoa* that had been sampled from the Canary and Balearic Islands [15,104]. In this work, the use of 3G8 mAb-MBs and 10C9 mAb-MBs separately instead of combined allowed discrimination between the CTX1B and CTX3C series of congeners in the samples. In this particular case, one advantage of immunosensing tools compared to CBA is that maitotoxin (MTX), a compound usually found in *Gambierdiscus* and *Fukuyoa* that interferes in CBA, is not recognized by the antibodies and does not cause any interference on CTX recognition. Campàs and co-authors explored the adaptation of this immunoassay to an integrated and portable device [105]. To avoid the use of magnets and MBs, the two capture antibodies were immobilized on carbon electrodes coated with multiwalled carbon nanotubes. The electrochemical measurements were performed with a miniaturized potentiostat connected to a smartphone. The system was validated using naturally contaminated fishes from Japan and Fiji, and good correlations were obtained with other techniques. This portable device has the potential to be applied to in situ analysis.

Optical biosensors, where the detection is based on variations in optical properties of light induced by biorecognition events (e.g., antibody-toxin interaction), have also been developed for analysis of TTXs. Several surface plasmon resonance (SPR) biosensors, which are label free, have been described [106,107,108]. The immobilization of TTX on SPR chips was typically achieved using SAMs, but Campbell and collaborators proposed the direct immobilization of TTX on carboxymethylated chips to simplify the experimental protocol and to shorten analysis time [109]. This biosensor was applied to the analysis of pufferfish and shellfish samples, demonstrating good sensing capabilities. Reverté and co-workers evaluated the use of a planar waveguide optic nanoarray biosensor with fluorescent detection for the determination of TTXs in pufferfish [110]. The signal was generated from the light transmitted through the waveguide leading to the excitation of the fluorophores used as labels. This optical biosensor was successfully applied to the analysis of *Lagocephalus lunaris* individuals from USA and *L. sceleratus* individuals from Greece.

### 3.4. Aptamer-Based Biosensors

In comparison with other fields such as biomedicine, the application of aptamers in toxin analysis and food safety is relatively scarce. Nevertheless, some recent studies make evident their potential in the development of biosensors for TTXs. Shao and co-workers were the first to obtain an aptamer against TTXs [111] and used it in the development of an assay [112]. In the absence of TTX, the aptamer hybridizes with its complementary sequence and an intercalator dye generates a fluorescent signal. In the presence of TTX, the aptamer acquires a tridimensional structure around the toxin that makes impossible the formation of dsDNA, and the signal decreases. Lan and co-authors used another dye that increases the fluorescent signal when the aptamer folds around the TTX. The assay demonstrated high sensitivity, being able to detect TTXs at levels as low as 0.1 nM. Additionally, the matrix interferences from human serum samples were evaluated, and a 100% recovery was obtained in the analysis of spiked samples [113]. The same strategy was recently applied to pufferfish samples [114]. Dou and colleagues designed an assay based on the fluorescence resonance energy transfer (FRET) phenomena to detect TTXs in shellfish samples [115]. Fluorescent nanoscale metal-organic frameworks (NMOFs) were used as an energy donor. The aptamer, modified with a fluorescent dye acting as an acceptor, was immobilized on the NOMFs. Upon the addition of TTX, the aptamer folded, shortening the distance between the acceptor and donor, enhancing the FRET, and increasing the signal produced by the dye.

Anti-TTX aptamers have also been used in the construction of hybrid assays combined with other biorecognition elements (Figure 5). Shkembi and collaborators developed a colorimetric sandwich assay for the detection of TTXs using an anti-TTX mAb for the capture and a labeled aptamer for signal development [116]. This hybrid assay was successfully applied to the analysis of a pufferfish of the species *L. sceleratus*.

Fomo and collaborators were the first to develop an electrochemical aptamer-based biosensor for the detection of TTXs [117]. Their strategy was based on the immobilization of the aptamer on a glassy carbon electrode coated with a conducting polyaniline layer. The interaction of the aptamer with the TTX modulates the electrochemical properties of the biosensor. Even though the impedimetric aptamer-based biosensor was able to detect TTX at levels as low as 0.19 ng/mL, analysis of seafood matrices is necessary to validate its applicability.

## 4. Analysis of Samples

The performance of a biosensor can suffer from interferences caused by the samples. These interferences are typically due to matrix compounds present in raw extracts, which have been co-extracted with the toxins, and may be responsible for under or overestimation of the toxin contents. Other factors such as the solvents used for the extraction may also affect the analysis performance. Therefore, sample processing and toxin extraction are not trivial and need to be carefully considered during the development of new detection approaches as well as in assessment of their applicability.

Toxins are not homogeneously distributed in seafood, and certain tissues are more susceptible to accumulating toxins than others. In fish, viscera (liver, gonads, and digestive tract) used to have higher CTXs or TTXs contents than, for example, muscle or skin [118]. The matrix composition of each tissue and how it affects the assay performance cannot be accurately predicted and depend on the fish individual, species, origin, size, and age. For example, liver is one of the most complex tissues to be analysed due to its composition in fatty acids, and the matrix interferences are usually more notorious in larger fishes. The matrix effects from liver can be overcome by removing fats with a liquid/liquid partition with organic solvents [92]. From a food safety point of view, the analysis of edible tissues, usually muscle, is a priority. Nevertheless, liver can be a suitable sentinel to identify toxins in natural samples and can be analysed in parallel, at least in research activities. Knowledge about TTXs contents in pufferfish tissues other than muscle may also be important during slicing for culinary purposes, since cross-contamination can occur. In monitoring programs, budgetary reasons may restrict the analysis to only one tissue per specimen.

After the development of a biosensor or other analytical system, validation is required. First, negative extracts are usually analysed to assess if the matrix compounds interfere on the assay. Even though we have previously mentioned that matrix effects are difficult to predict, their evaluation can contribute to assure proper biosensor performance. In some detection approaches, analysis of serial dilutions of the extract may provide the highest matrix concentration that can be used with no interference. However, when diluting a sample to reduce the matrix concentration, the toxin contents also decrease, compromising toxin detection. This can be a limiting step when trying to detect TTXs in shellfish, as it usually contains lower toxin amounts than pufferfish. Second, toxin-spiked homogenates or extracts should be analysed. Whereas the analysis of spiked extracts provides information about the effect of the matrix on toxin detection, the analysis of spiked homogenates involves the prior toxin extraction process and, therefore, extraction recovery is also evaluated. From spiking experiments, correction factors can be established, which will contribute to provide more accurate toxin quantifications.

The elimination of interfering matrix compounds using different clean-up procedures may increase the sensitivity and reliability of bioanalytical tools. Apart from commercial SPE cartridges, new biotechnological materials can be used. Campàs and co-workers used cyclodextrin polymers to purify fish extracts containing CTXs for its subsequent analysis with CBA [119]. This clean-up strategy allowed the exposure of cells to high matrix concentrations (fish flesh equivalent concentrations as high as 400 mg/mL) and provided at least 4.6- and 3.0-fold higher sensitivities to the assay for *Variola louti* and *S. dumerili*, respectively. The use of MBs in clean-up and preconcentration strategies, instead of as immobilization supports, has also been explored. Campbell and collaborators conjugated anti-STX mAb to MBs and used the immunoconjugates to purify naturally contaminated mussels [120]. The use of immunoaffinity allowed the capture of STX from complex matrices for subsequent high-performance liquid chromatography (HPLC) analysis with lower matrix effects. It seems reasonable to think that this strategy could be applied to the purification of samples containing TTXs or CTXs, provided that appropriate antibodies are available, and the cost is affordable. The combination of these clean-up strategies with biosensors instead of assays is still a pending task.

Biosensors can be tailor-designed to be used as screening tools or as precise quantification methods. In screening, action limits (or action regions, to be safe) close to the regulatory levels are established, and the result of the analysis will indicate if the sample contains toxin contents below or above a threshold. Usually, when the result is near the action limit of inside the safety action region, the analysis should be repeated with a confirmatory technique. Screening can also provide a yes/no presence response. In quantitative approaches, extracts are usually analysed and quantified at different dilutions. It may happen that at high matrix concentrations, quantifications are under or overestimated, and this effect may disappear at lower matrix concentrations, when the toxin contents tend to stabilize.

## 5. Comparison among Techniques

The validation of biosensors and bioanalytical assays requires comparison of the quantifications of toxins in samples with those obtained with other techniques. Ideally, the new smart bioanalytical tools should provide results similar to those obtained with the reference analytical techniques, if available. However, differences in toxin contents among techniques can be observed due to the different detection principles, making comparison not an easy task.

Cell-based approaches provide a composite response related to the toxicity of the sample, which includes the effect of all toxic congeners present in that sample. Therefore, toxin quantifications with CBAs or CBBs are expressed as equivalents of toxicity of the parent analogue (normally the most toxic). Liquid chromatography coupled to mass spectrometry, a technique based on a unique and precise structural recognition and widely used in the analysis of marine toxins, provides individual contents of each analogue. The application of the TEF of each analogue to individual quantifications from LC-MS/MS gives as a result the toxin equivalents of a reference toxin, which can provide a better estimation of the toxicity of the sample and, therefore, it is more interesting from a consumer and health protection point of view. LC-MS/MS can provide the multi-toxin profiles of samples, information that other techniques cannot.

Immunosensing strategies also provide a composite response but, unlike CBA, this response is related to structure. In a similar way, the application of CRFs to LC-MS/MS quantifications helps to better compare the contents provided with these two techniques and enables the validation of immunosensing tools. This comparison has been made in the analysis of pufferfish samples with ELISA [92,94], optical biosensors [92,110], and electrochemical biosensors [93]. It is important to mention that CRFs may depend on the immunosensing format and detection technique [92,121], which has sense considering that different immunoassay/immunosensor configurations can favor or hinder the interaction of the antibody with its target. Regarding aptamers and receptors, the availability of CRFs for different toxin analogues would also be interesting. However, their elucidation is still a pending task. As previously mentioned, in an ideal situation, CRFs should correlate with TEFs. In this way, the toxin contents provided by structural approaches would provide useful information to protect seafood consumers. It is important to add that the scarcity of TTX and CTX congeners makes the establishment of CRFs and TEFs highly complex. Nevertheless, progress is being made in this direction.

Another parameter that may affect the comparison among techniques is the matrix effect since matrix compounds may differently affect the recognition and detection steps, depending on the analytical technique. Also important is the limit of detection, as samples considered negative with a particular technique may be positive with one more sensitive.

## 6. Miniaturization and Automation

Usually, samples need to be transported from the sampling point to the laboratory to be analysed, which involves a time lag between sampling and data generation. When dealing with fresh food products, this time lag could pose a problem because the analysis process may delay the release of the product to the market, compromising its quality. Time is even more critical when the product has already reached the market. Furthermore, rapid analysis is essential to avoid long production shutdowns and consequent economic losses. Bioanalytical assays and biosensors for toxin monitoring are gaining an increasing interest in food safety because of their potential to be used as point-of-care (POC) devices in resource-limited settings. However, there are several factors that need to be adjusted to bring analysis of toxins to the field.

One of the critical issues in the development of POC devices for the detection of toxins is sample processing. Generally, POC devices are easily applied to samples that do not require any pre-treatment (e.g., blood, urine, or saliva). However, analysis of seafood requires toxin extraction, and to simplify the extraction protocols to be performed out of the laboratory is not an easy task. Nevertheless, some advances have been made in this field. The use of manual rollers and filtration bags as alternatives to blenders and centrifuges has been tested and, in fact, is included in some commercial kits. Additionally, portable bead beaters have been used to lyse *Gambierdiscus* cells [20] and, although the purpose is different, this strategy could be tested for the extraction of toxins from tissues.

Another important issue of POC devices is whether the resulting information is qualitative or quantitative. Quantitative measurements certainly require a calibration curve, whereas for qualitative measurements, only positive and negative controls may be enough. Many commercial kits already include several toxin standards at different concentrations to be tested simultaneously to the sample. Regarding biosensors, the use of electrode arrays and multiplexer potentiostats also enable the analysis of several samples and/or calibration standards at the same time. In POC devices, calibrations, and controls should be integrated into the tools, and this may complicate their design and development.

The size and portability of the instrumentation required in biosensing are other factors that need to be assessed for in situ analysis. As previously mentioned, electrode arrays and multiplexed potentiostats for the analysis of several samples at the same time are available as well as portable potentiostats. In fact, potentiostats of the size of a pen drive, which can be connected to smartphones, are available and are ideal for the development of POC devices. In a recent study from Campàs and co-authors, such a device is used for analysis of CTXs in fishes [105]. This is a clear example of the portability and integration of biosensing systems for in situ analysis. In the next few years, lateral flow assays, such the those that screen for SARS-CoV-2, will certainly have a role in the analysis of CTXs and TTXs.

## 7. Quality Control

To ensure that the data produced by new analytical tools are fit for their intended purpose, internal quality controls should be undertaken. Internal quality controls are an important determinant of the quality of analytical data and recognized as such by accreditation agencies. Usually, certified reference materials (CRMs) should be included in the analytical sequence. Regarding CTXs, no CRMs are commercially available. Nevertheless, non-certified CTX1B, CTX3C, and maitotoxin standards can be bought from FUJIFILM Wako Pure Chemical Corporation [122] and gambierone and 44-methyl-gambierone are commercialized by CIFGA S.A. [123]. Regarding TTXs, there are two commercially available CRMs: one containing only TTX from the National Research Council Canada [124] and another one containing a mixture of TTX and two TTX analogues from CIFGA S.A [123]. Non-certified TTX standards are also commercially available. Additionally, non-certified standards can also be obtained from research institutions. Due to the lack or scarcity of CRMs, these non-certified standards can be used for calibrations and as internal quality controls. These control materials should be traceable at least to a material of guaranteed purity or other well characterized material. Additionally, issues such as instrument settings, the operators and their competence, the reagents’ stability, and the laboratory environment may affect the quality of results, including precision, accuracy, and repeatability among other parameters, and therefore need to be controlled to assure the quality of the data. The lack or scarcity of standards and CRMs makes difficult the harmonization of analytical methods as well as collaborative validations and proficiency tests for external quality control.

## 8. Conclusions

The biosensors and other smart bioanalytical systems developed until now for the detection of CTXs and TTXs have proven to be useful for the analysis of these toxins in fish and shellfish. However, it is important to mention that each one of them has its own strengths and weaknesses (Table 1). Cell-based approaches provide a toxicological evaluation of the sample. However, the use of CBAs may suffer from certain variability. On the other hand, biosensing devices for electrophysiological recordings are expensive. Regarding receptors, the work undertaken in this field is scarce, probably being limited by the need to obtain synaptosomes from primary cultures, the filtration and centrifugation steps, and the type of labels required, usually radioactive compounds. Nevertheless, fluorescence labels are gaining importance. The number of immunosensing systems is higher, being antibodies the biorecognition elements more commonly used in the development of biosensors for the detection of CTXs and TTXs. The high specificity, sensitivity, stability, and biocompatibility with most sensing supports make antibodies ideal for biosensing. The use of aptamers is also interesting due to their high specificity, sensitivity, and versatility, but the need for high amounts of target may hinder their production. 

Analysis of samples with these novel biosensing tools and comparison with other techniques have contributed to their validation, but some research gaps still need to be filled. The lack or scarcity of CRMs, the complex toxin chemical structures, the large families of analogues, the multi-toxin profiles, the wide range of seafood species, the difficulties in sample preparation, and the extremely low limits of detection required are some of the challenges to be solved in the next few years. Nevertheless, the results obtained so far are promising and underline the high performance of these bioanalytical tools, which certainly have the potential to be integrated in research activities and monitoring programs.

## Figures and Tables

**Figure 1 foods-12-02043-f001:**
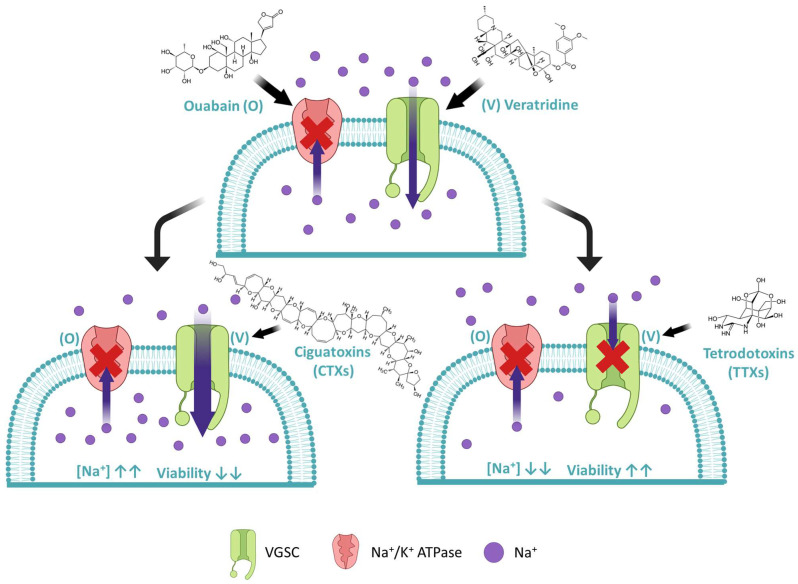
Schematic representation of the effect of ciguatoxins (CTXs) and tetrodotoxins (TTXs) on the sodium flow dynamics through the membrane of Neuro-2A cells treated with ouabain (O) and veratridine (V).

**Figure 2 foods-12-02043-f002:**
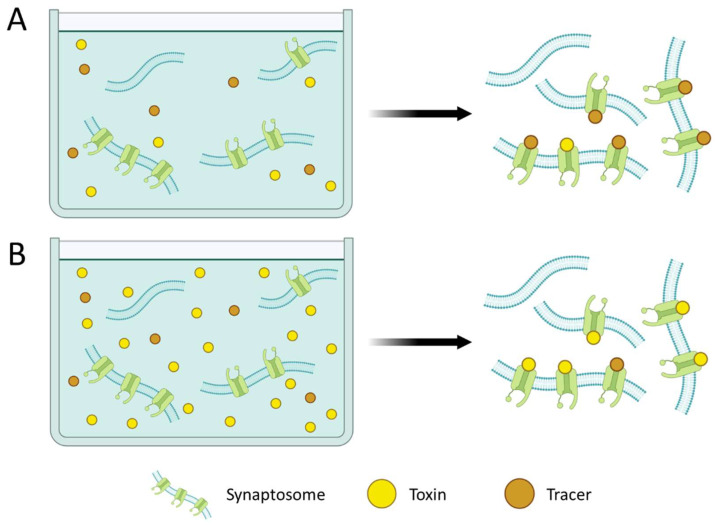
Schematic representation of a competitive receptor binding assay (RBA) applied to samples with low toxin contents (**A**) and to samples with high toxin contents (**B**). The lower the toxin content is in the sample, the more the labeled ligand is bound to the receptor, and the higher the signal is, and vice versa.

**Figure 3 foods-12-02043-f003:**
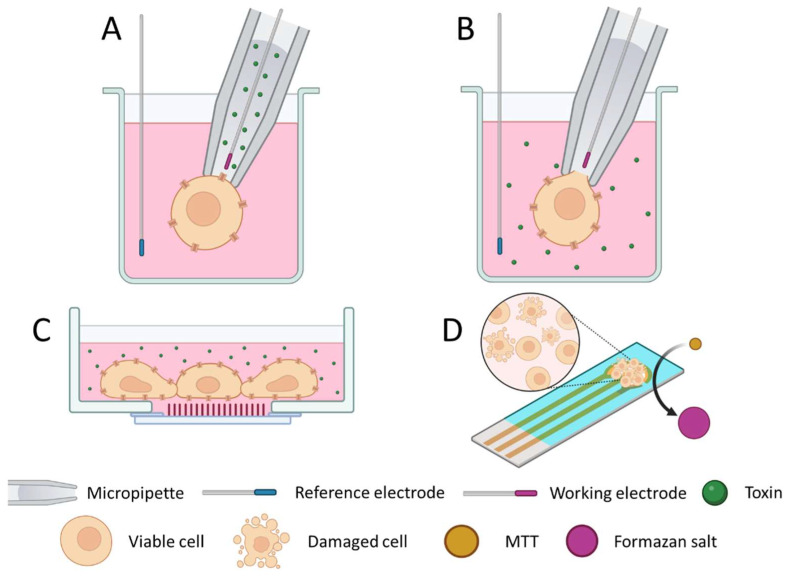
Cell-based biosensor (CBB) approaches based on electrophysiology (**A**–**C**) and cell viability evaluation (**D**) for the detection of ciguatoxins (CTXs) and tetrodotoxins (TTXs). Cell-attached patch clamp (**A**), whole-cell patch clamp (**B**), microelectrode array (MEA) well (**C**), and cells immobilized on electrodes (**D**).

**Figure 4 foods-12-02043-f004:**
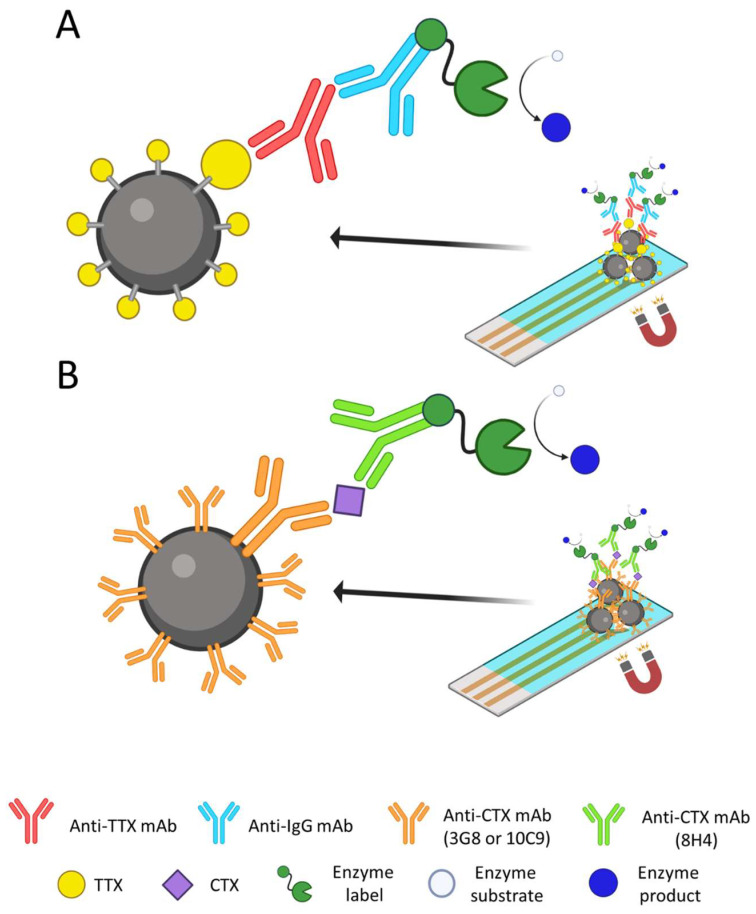
Schematic representation of immunosensors for tetrodotoxins (TTXs) and ciguatoxins (CTXs). The immunosensor for TTXs is based on a competition between the toxin in the sample and the immobilized toxin for binding to the anti-TTX mAb (**A**). The immunosensor for CTXs is based on a sandwich configuration using two anti-CTX mAbs (**B**).

**Figure 5 foods-12-02043-f005:**
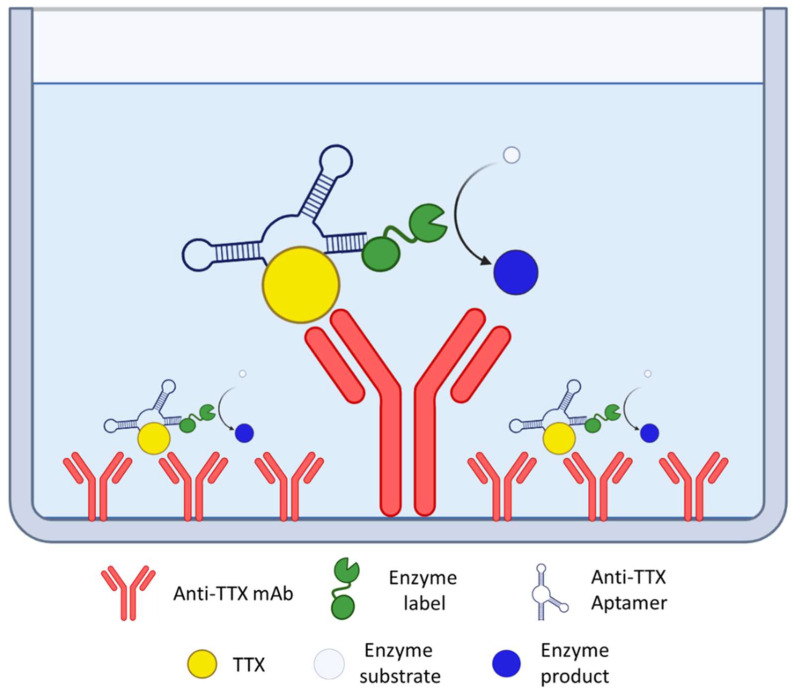
Schematic representation of a hybrid sandwich aptamer-based assay for TTXs.

**Table 1 foods-12-02043-t001:** Comparison of performance parameters of different biosensors for the detection of ciguatoxins (CTXs) and tetrodotoxins (TTXs).

Biosensor Type	Cell-Based Biosensors (CBBs)	Receptor Binding Biosensors (RBBs)	Immunosensors	Aptamer-Based Biosensors
Biorecognition element	Cells (primary cultures or immortal cell lines)	Receptors (synaptosomes)	Antibodies (mono-clonal or polyclonal)	Aptamers (RNA or DNA)
Approach	Toxicological	Structural	Structural	Structural
Supports	Micropipettes, patch clamp chips, micro-electrode arrays (MEAs), electrodes	RBB not yet developed; only receptor-binding assays (RBAs) on microtiter plates	Electodes, magnetic beads, surface plasmon resonance (SPR) chips, planar waveguide nanoarray chips	Electrodes
Sensitivity	High	Medium	High	High
Toxins	CTXs and TTXs	Only RBAs for CTXs and TTXs	CTXs and TTXs	TTXs
Samples	Fishes (CTXs and TTXs) and spiked mussels (TTXs)	Only RBAs for fishes (CTXs) and microalgae (CTXs)	Fishes (CTXs and TTXs), microalgae (CTXs), and shellfish (TTXs)	Only aptamer-based assay for fishes (TTXs) and shellfish (TTXs)

## Data Availability

The data presented in this study are available in the article.

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
