# Peer review of "Detection of Ciguatoxins and Tetrodotoxins in Seafood with Biosensors and Other Smart Bioanalytical Systems"

_foods, 2023, doi:10.3390/foods12102043_

Round 1

Reviewer 1 Report

The submitted manuscript is comprehensive and corresponds to the topic the authors wish to present in their review paper.

The authors deal with a very interesting and significant topic.

The topic of the manuscript for which the authors decided has a great scientific contribution, and great practical importance.

The manuscript is well organized.

The parts of the manuscript are, apart from the Introduction, "2. Biorecognition Elements", "3. Biosensors and other Smart Bioanalytical Systems", "4. Analysis of samples", "5. Comparison among techniques", "6. Miniaturization and automation" and "5 . Conclusions".

"5. Conclusions" (Line ) should be "7. Conclusions".

In my opinion, a minor flaw in this manuscript is that nowhere (in parts 4, 5 and/or 6) does it contain a description of the internal quality control of the results. It has to do with the procedures and validation procedures mentioned in parts 4 and 5).

This should be related to the resources used (equipment and its measurement accuracy, CRM, RM or control samples used, personnel/performers and their competence.

Are there any PT activities/schemes available for this purpose?

the analytical part, methodologies, used equipment, etc., are all very new, so a comprehensive internal-external control is needed.

I am not fully qualified to judge the quality of English in this paper.

However, I could notice that "Minor editing of English language required".

Author Response

The submitted manuscript is comprehensive and corresponds to the topic the authors wish to present in their review paper.

The authors deal with a very interesting and significant topic.

The topic of the manuscript for which the authors decided has a great scientific contribution, and great practical importance.

The manuscript is well organized.

Thanks for your comments.

The parts of the manuscript are, apart from the Introduction, "2. Biorecognition Elements", "3. Biosensors and other Smart Bioanalytical Systems", "4. Analysis of samples", "5. Comparison among techniques", "6. Miniaturization and automation" and "5. Conclusions".

"5. Conclusions" should be "7. Conclusions".

Sorry for the mistake. We have corrected it (now it is section 8).

In my opinion, a minor flaw in this manuscript is that nowhere (in parts 4, 5 and/or 6) does it contain a description of the internal quality control of the results. It has to do with the procedures and validation procedures mentioned in parts 4 and 5).

This should be related to the resources used (equipment and its measurement accuracy, CRM, RM or control samples used, personnel/performers and their competence.

Are there any PT activities/schemes available for this purpose?

The analytical part, methodologies, used equipment, etc., are all very new, so a comprehensive internal-external control is needed.

Techniques are too new, as the reviewer mentions, to be accredited and implemented in monitoring programs right now (although some of them are promising!). Nevertheless, these issues are very pertinent, and they have now been mentioned in a new section (section 7).

Comments on the Quality of English Language I am not fully qualified to judge the quality of English in this paper. However, I could notice that "Minor editing of English language required".

English language has been revised and minor corrections have been made.

Reviewer 2 Report

1.        The period of publications reviewed in this article should be specified.

2.        The growth curve showing the number of publications over the past 5 or 10 years should be plotted to emphasize the importance of a review on this topic.

3.        A new subsection (preferably subsection 2) should be included to summarize the classical traditional methods of analysis (besides biosensors) and their shortcomings.

4.        A table should be included to summarize the toxin detection methods discussed in various subsections highlighting the method name, recognition element, sensitivity, linearity, detection/quantitation limit, application to real/synthetic samples, and limitations for each method discussed.

5.        The conclusion section should also contain future perspectives highlighting the possible research gaps need to be filled for efficient detection of ciguatoxins and tetrodotoxins.

Minor editing of English language required

Author Response

  1. The period of publications reviewed in this article should be specified.
  2. The growth curve showing the number of publications over the past 5 or 10 years should be plotted to emphasize the importance of a review on this topic.

This is a combined answer for both questions. This article reviews all the existing biosensing systems for ciguatoxins and tetrodotoxins regardless the period. The number of publications is still scarce, although the field is promising, and the development of bioanalytical tools for seafood safety is gaining interest. To meet the reviewer requirements, we have added this sentence: “In the last 15 years, the number of publications with regards to new analytical methods for marine toxins has reached 200 per year, and the citations have increased from 5,000 to more than 11,000.”

  1. A new subsection (preferably subsection 2) should be included to summarize the classical traditional methods of analysis (besides biosensors) and their shortcomings.

Classical traditional methods, with their advantages and limitations, have been summarised. To not deviate from the purpose of the review, this paragraph has been placed in section 1 (before starting the sections about alternative methods).

  1. A table should be included to summarize the toxin detection methods discussed in various subsections highlighting the method name, recognition element, sensitivity, linearity, detection/quantitation limit, application to real/synthetic samples, and limitations for each method discussed.

A table has been included summarising the performance parameters of the biosensors. The parameters are not exactly those suggested by the reviewer, due to the scarcity of some biosensors and the fact that most of them have not been fully developed and/or validated. Therefore, a more critical table has been constructed, summarising advantages and limitations and empathizing the critical issues.

  1. The conclusion section should also contain future perspectives highlighting the possible research gaps need to be filled for efficient detection of ciguatoxins and tetrodotoxins.

The research gaps and future perspectives have now been mentioned “…some research gaps still need to be filled. The lack or scarcity of CRMs, the complex toxin chemical structures, the large families of analogues, the multi-toxin profiles, the wide range of seafood species, the difficulties in sample preparation and the extremely low limits of detection required are some of the challenges to be solved in the next few years.”

Comments on the Quality of English Language Minor editing of English language required.

English language has been revised and minor corrections have been made.

Round 2

Reviewer 2 Report

The authors have satisfactorily addressed all the comments raised by reviewers and therefore I recommend acceptance of this article for publication in Foods.

Minor editing of English language required